# Interactive Effects of Copper Pipe, Stagnation, Corrosion Control, and Disinfectant Residual Influenced Reduction of *Legionella pneumophila* during Simulations of the Flint Water Crisis

**DOI:** 10.3390/pathogens9090730

**Published:** 2020-09-04

**Authors:** Rebekah L. Martin, Owen R. Strom, Amy Pruden, Marc A. Edwards

**Affiliations:** 1Department of Civil and Environmental Engineering, Virginia Tech, 418 Durham Hall, Blacksburg, VA 24061, USA; hrebek1@vt.edu (R.L.M.); apruden@vt.edu (A.P.); 2Department of Civil and Environmental Engineering, Virginia Military Institute, Lexington, VA 24450, USA; 3Elson S. Floyd College of Medicine, Washington State University, Spokane, WA 99202, USA; owen_strom@wsu.edu

**Keywords:** flint, *Legionella pneumophila*, copper, PEX, iron, chlorine, premise plumbing

## Abstract

Flint, MI experienced two outbreaks of Legionnaires’ Disease (LD) during the summers of 2014 and 2015, coinciding with use of Flint River as a drinking water source without corrosion control. Using simulated distribution systems (SDSs) followed by stagnant simulated premise (i.e., building) plumbing reactors (SPPRs) containing cross-linked polyethylene (PEX) or copper pipe, we reproduced trends in water chemistry and *Legionella* proliferation observed in the field when Flint River versus Detroit water were used before, during, and after the outbreak. Specifically, due to high chlorine demand in the SDSs, SPPRs with treated Flint River water were chlorine deficient and had elevated *L. pneumophila* numbers in the PEX condition. SPPRs with Detroit water, which had lower chlorine demand and higher residual chlorine, lost all culturable *L. pneumophila* within two months. *L. pneumophila* also diminished more rapidly with time in Flint River SPPRs with copper pipe, presumably due to the bacteriostatic properties of elevated copper concentrations caused by lack of corrosion control and stagnation. This study confirms hypothesized mechanisms by which the switch in water chemistry, pipe materials, and different flow patterns in Flint premise plumbing may have contributed to observed LD outbreak patterns.

## 1. Introduction

The Flint Water Crisis began when the City of Flint, Michigan switched from purchasing its long-term Detroit municipal water supply (sourced from Lake Huron) to the local Flint River in April 2014. Although the Flint River water was predictably much more corrosive than Detroit water, no federally-mandated corrosion control program was implemented, resulting in rampant corrosion of lead plumbing and iron water mains, low chlorine residuals, elevated bacteria, and high levels of lead [1,2,3,4]. During the summers of 2014 and 2015, Flint also experienced two outbreaks of Legionnaires’ Disease (LD), with 91 cases and 12 deaths documented in Genesee County (the county which Flint is located), compared to the 6–13 cases per year and no deaths during 2009–2013 while on Detroit water [1,3,5,6]. The period of Flint River water use was characterized by high levels of LD incidence [6] and *L. pneumophila* gene marker levels [1] associated with large buildings. However, lower LD incidence associated with residential exposure was noted from August 2015 onwards and our sampling during this period revealed undetectable or very low levels of *L. pneumophila* in residential plumbing [1,3]. While temporal associations between the switch in water supply with reduced levels of chlorine, high levels of iron, elevated temperature for Flint River water, and the resulting outbreak of LD were predictable based on prior work [7,8,9,10], and duly noted for the Flint outbreak [1,3,5], precise patterns of the outbreak in relation to large health care facilities versus residential single family homes are still the subject of scientific and public interest [5,11].

We recently examined *L. pneumophila* growth in simulated glass water heaters with either cross-linked polyethylene (PEX) or copper pipe, Detroit tap water (sourced from Lake Huron) or treated Flint River water, and an initial influent pH 7.3 with continuous mixing representing hot water recirculation often present in large buildings [12]. After one year, very high levels of *L. pneumophila* (2.6–3.0 log CFU/mL) were observed in all treated Flint River water conditions with PEX pipe and with copper pipe when there was even low levels of phosphate corrosion control present (2.9 log_10_ CFU/mL). By contrast, all treated Flint River water conditions with copper pipe and no phosphate corrosion control had 1–2 log lower levels of *L. pneumophila*, likely due to biotoxicity of copper, as directly evidence by an inverse correlation (R^2^ = 0.85–0.95) between *L. pneumophila* and measured soluble copper or Cu^+2^ [12].

Premise plumbing, and its resident microbiome, is highly sensitive to variation in pipe material, pH, disinfection conditions, and stagnation. All of these factors can produce important synergistic or antagonistic effects [13]. For example, a 0.5-unit higher pH in the influent, would be expected to reduce bacteriostatic effects of copper pipe on resident microbes due to reduced concentrations of Cu^+2^ and soluble copper [14,15]. Likewise, complete stagnation has sometimes been associated with greatly reduced growth of *Legionella* versus either completely-mixed (i.e., water recirculation) or more frequent flow [16,17,18,19] conditions, whereas the opposite effect is expected if the water has high levels of disinfectant or very high temperature [20]. The presence of free chlorine has also been associated with lower levels of *Legionella* in general [21,22,23] and with LD incidence, in particular, during the Flint Water Crisis [1,3,5]. Thus, it is of interest to evaluate the extent to which recent phenomena observed to be at play in Flint’s premise plumbing [12] hold true under a broader range of relevant conditions. Specifically, conditions with a higher influent pH (7.8–8.5), stagnation (which is more common in residences than large building hot water systems), and the free chlorine levels representative of before, during, and after the Flint Water Crisis.

Here, we evaluated how the corrosive treated Flint River water and the less corrosive Detroit tap water interacted with unlined iron water mains, and then subsequently with the premise plumbing pipe materials into which that water flowed, to influence levels of disinfectant and the propensity for *Legionella* growth (Figure 1). The overarching hypothesis was that the lack of corrosion control of Flint River water would cause higher iron and lower chlorine after contact with unlined iron pipe mains (Figure 1), creating conditions less likely to disinfect *Legionella* when this water flowed into stagnant PEX plumbing. Copper pipe, which has the potential to either catalyze chlorine decay and thereby hinder disinfection [24,25,26], or release antimicrobial soluble copper ions and enhance disinfection, was compared to a control with PEX pipe for all conditions. The expectation was that *Legionella* would survive best in treated Flint River water with PEX versus copper due to little or no chlorine delivery, but that the converse would be true in Detroit tap water if copper pipe catalyzed chlorine decay and the high levels of corrosion control would virtually eliminate passive disinfection by copper (Figure 1). This study provides important insight into interactive effects of water chemistry and pipe material in affecting the trajectory of community-wide LD outbreak.

## 2. Materials and Methods

### 2.1. Source Water Treatment

Raw water was directly collected from the Flint River at GPS coordinates 43.018230, −83.693944. Lake Huron-sourced drinking water (Detroit tap water) was collected after > 5 min flushing from the tap of a residential Flint home. Raw Flint River water and Detroit tap water were both collected on 18 August 2016, 21 September 2016, 11 October 2016, 21 November 2016, and 27 January 2017 and express shipped to Blacksburg, Virginia in 30-L containers. Additional raw Flint River water samples were collected and shipped express on 8 February 2017 and 11 March 2017. All collected water was stored at 4 °C prior to preparation for experiments.

Water treatments applied to raw Flint River water during the crisis were simulated in the laboratory. These included 56 mg/L ferric chloride for coagulation, 10 min of stirring for flocculation, 159 mg/L lime as Ca(OH)_2_ for softening, followed by another 15 min of flocculation. The water was subsequently settled for 4 h and filtered through a column of glass wool to simulate sand filtration. Water treated in this manner was designated as “treated Flint River” water (Figure 2). Working stocks (10–20 L) of treated Flint River water and Detroit tap water were stored at room temperature (23 °C) until the supply was exhausted.

### 2.2. Simulated Distribution Systems: Chlorination and Corrosion

Six SDS conditions served to reproduce distributed waters that either occurred (conditions designated in **bold** font) under conditions relevant to the Flint Water Crisis or its aftermath or hypothetical scenarios if corrosion control had been implemented or if water had not flowed through unlined iron pipe (conditions designated in *italics*) (Figure 2). In five of the six conditions, the practical influence of unlined iron distribution system pipe was simulated by addition of an iron wire to flasks mixing each water for 3 h. Treated Flint River water conditions included a condition with the omission of phosphate corrosion control (as was the case during the crisis) (**FR**), a hypothetical condition if 1 mg/L as PO_4_-P orthophosphate corrosion control had been implemented (*FR-CC*), and a condition without any phosphate corrosion control or iron corrosion (i.e., no iron wire) (*FR-NoFe*) representing some sections of Flint served by newer concrete lined iron or PVC distribution system pipe during the crisis. Detroit tap water conditions examined the pre-crisis effect of Lake Huron-sourced water with lower distribution system temperature (**DET-Cold**) containing 2.5 mg/L orthophosphate PO_4_-P, the post-crisis water with enhanced doses of chlorine and additional phosphate (3.5 mg/L chlorine and 4.0 mg/L orthophosphate) to assist with system recovery once Flint switched back to Detroit-sourced water (**DET-Enhanced**), and a hypothetical condition if normal Detroit distribution water with 2.5 mg/L orthophosphate had been as warm as treated Flint River water during summer months (*DET*).

### 2.3. General SDSs Water Preparation

Sodium hypochlorite (10% diluted Clorox™ bleach, the Clorox Company, Oakland, CA, USA) was added to 330 mL of each water condition until an initial stable target of 3 mg/L free chlorine residual was obtained (the only exception being a higher residual of 3.5 mg/L in **DET-Enhanced**), followed by the SDSs in 500 mL glass flasks containing magnetic stir bars and mixing 400 rpm for three hours. In all conditions, except *FR-NoFe*, the presence of iron pipe was simulated in the SDS with a 12 cm length of 99% 2 mm diameter iron wire (approximately, 7.6 cm^2^ Fe surface per liter of water) and orthophosphate was added to achieve corrosion control targets of 1 (*FR-CC*), 2.5 (*DET*, **DET-Cold**), or 4.0 mg/L (**DET-Enhanced**).

### 2.4. Premise Plumbing

#### 2.4.1. Simulated Premise Plumbing Reactors (SPPRs)

Following the SDS step, waters were transferred to 100 mL borosilicate glass bottles (36 total) designed to simulate changes occurring in water as it ages in premise plumbing (SPPR, Figure 2). Each SPPR was equipped with either eight pieces of 20 mm × 10 mm cross-sectional PEX (n = 18) or solid copper (n = 18) pipe material. Pipe coupons had been aged in the bottles for six years in prior experiments, described elsewhere [8,9,10,27], which provided a benefit of well-aged premise plumbing pipe materials and mature biofilms at the start of the experiment.

#### 2.4.2. Initializing the SPPRs

All 36 SPPRs were conditioned prior to the experiment, by dosing a homogeneous aliquot of reactor effluents according to pipe material, followed by an acclimation phase of 50% water volume changes with treated Flint River water every three days for 101 days. This water change frequency and volume simulated a low use, high-stagnation condition considered to be conducive to *Legionella* growth in premise plumbing [8]. On Day 14, the SPPRs were inoculated with three environmental *L. pneumophila* isolates from Flint, MI buildings at a total concentration of 10,000 colony forming units per milliliter (CFU/mL). The inoculum was composed of a mixture of *L. pneumophila* serogroup 1 as well as two non-serogroup 1 isolates. Inoculum concentration was determined by optical density readings of *L. pneumophila* colonies scraped from agar plates, resuspended in Nanopure water, and measured at 600 nm using a 4500 HACH spectrophotometer (Hach Company, Loveland, CO, USA).

To avoid introduction of *Legionella* spp. that may have been present in the water shipments once the experiment was in progress, effluent SDS waters were monitored for survival of culturable *Legionella* prior to their addition to corresponding SPPRs. In no case was detectable culturable *Legionella* present after chlorination of the water and incubation in the SDSs.

#### 2.4.3. Water Changes with SDS Conditions

Following inoculation and a 101-day conditioning period with treated Flint River water, 50% water changes were performed every 3–4 days for 175 days. SPPRs were reproducibly inverted five times for each water change to resuspend any settled material, and 50% of the volume was decanted and replaced with water from one of the six SDS conditions (Figure 2). Each of the six SDS conditions were tested in triplicate copper or PEX SPPRs. Reactors were incubated under static conditions at 37 °C between water changes. Thus, the experimental design included 6 SDS conditions × 2 pipe materials × 3 replicates = 36 total SPPRs.

Culturable *L. pneumophila* were enumerated as colony forming units per deciliter (CFU/mL) on Buffered Charcoal Yeast Extract (BYCE) agar (Remel, Lenexa, KS, USA) supplemented with 3 g/L glycine, 0.4 g/L L-cysteine, 80,000 units/L of polymyxin B sulfate, 0.001 g/L vancomycin, and 0.08 g/L cycloheximide. Initially, water was directly taken from SPPRs and plated onto BYCE; however, once CFUs dropped below detection of direct plating of 1 mL, 50 mL of effluent SPPR water was filter concentrated using 0.22 μM pore size mixed-cellulose ester membranes (Millipore, Billerica, MA, USA) and resuspended in 5 mL of Nanopure water prior to plating (1 mL). Water from each reactor was plated in triplicate. Plates were incubated at 37 °C for 5 days, after which *L. pneumophila* colonies were counted and CFU/mL were calculated. Direct plates with 0.02 CFU/mL were considered below detection. When no *L. pneumophila* colonies were detected from 50 mL concentrates, counts were considered below detection, resulting in a detection limit of 0.001 CFU/mL.

### 2.5. Culture Confirmation

To gain insight into the types of *Legionella* that persisted through the experiment, colonies visually determined as *Legionella* and non-*Legionella* species were picked from plates after 5 days of incubation at 37 °C for polymerase chain reaction confirmation. Polymerase chain reaction was used to confirm *Legionella* spp. (i.e., genus), *L. pneumophila*, and serogroup 1 using established primers and protocols [28,29].

### 2.6. Water Quality Analyses

Influent SDSs, effluent SDSs (influent SPPRs), and effluent SPPR waters were analyzed on Days 0, 9, 20, 72, 87, 126, 131, and 153. Inorganics, including dissolved and particulate iron and copper, were measured by inductively coupled plasma–mass spectrometry (ICP-MS) following 2% acidification with nitric acid. Total organic carbon (TOC) was measured according to standard method 5310 C using a persulfate-ultraviolet detection by a Sievers Model 5300 C (General Electric Company, Boston, MA, USA). pH was measured using an Oakton 110 series meter (Cole Parmer, Count Vernon Hills, IL, USA).

Free chlorine was measured using a 4500 HACH spectrophotometer (Loveland, CO) according to 4500-Cl standard method. To examine the kinetics of chlorine in the various water conditions used in this study, chlorine decay tests were performed on source waters (treated Flint River water and Detroit tap water) in non-reactive glass containers, on SDS water conditions with iron wire according to the experimental design (Figure 2), and after the SDSs waters were added to the SPPRs.

### 2.7. Data Analysis

Statistical tests were performed using R Studio (Version 1.0.153). A Shapiro–Wilk normality test was performed and none of the data were normally distributed. Arithmetic means were calculated for displaying results due to the high proportion of non-detect values in the dataset. Wilcoxon rank sum and Kruskal–Wallis rank sum tests with post-hoc Tukey tests were performed to determine statistical correlations. Wilcoxon tests were used for *Legionella* culture data (log transformed), pipe material, iron, and copper data, whereas the Kruskal–Wallis test was used to determine significance of chlorine data between SPPRs. Significance was set at a *p* value ≤ 0.05.

## 3. Results and Discussion

### 3.1. Simulated Treatment and Distribution Reproduced Key Factors of Pre-, During-, and Post-Crisis Flint Water

#### 3.1.1. Treated Source Waters Employed in this Experiment

To recreate water quality conditions in Flint, influent water conditions were simulated by treating raw Flint River water in the lab and collecting Lake Huron-sourced water from a well-flushed tap in Flint post crisis (Detroit tap water). The unaltered pH of treated Flint River water ranged between 7.84 and 8.57, while Detroit tap water ranged from 7.96 to 8.06, which recreated the stable pH observed when Flint was using Detroit water and the more variable pH when using Flint River water in 2014 [3,12].

The source water was added in 300-mL aliquots to six glass flasks (3 with Detroit tap water, 3 with treated Flint River water) with iron wire and mixed for 3 h to simulate six different conditions in distribution systems (SDSs). Just prior to being added to the SDSs, the source waters (treated Flint River water and Detroit tap water) were chlorinated, achieving an initial disinfectant residual of 3.10 mg/L Cl_2_ (Table 1 section B). Additional chlorine was added to only the **DET-Enhanced** SDS condition to achieve a higher average initial residual of 3.80 ± 0.19 mg/L (Table 1 section B). The possible short-term role of cooler temperature during distribution while on Detroit tap water was tested in this work with the **DET-Cold** SDS condition, held at an average of 18.3 ± 1.4 °C compared to an average 21.8 ± 1.3 °C of the other five SDS conditions (**FR**, *FR-NoFe*, *FR-CC*, *DET*, **DET-Enhanced**) (Table 1 section B,C). This ~3 °C difference served to recreate the reported average summer water temperature of 19.9 ± 2.24 °C (pre-crisis, Detroit) and 22.6 ± 2.14 °C (during crisis, Flint River) (Table 1 section A) [3].

#### 3.1.2. SDSs Chlorine

The effluent water collected following the 3-h SDSs reaction time (Figure 2) successfully replicated known trends in chlorine residuals observed in the Flint water distribution system before, during, and after the water crisis. To assess inherent chlorine demand prior to the SDSs step, treated Flint River and Detroit tap waters were aliquoted to non-reactive glass containers without iron. The chlorine residual in treated Flint River water dropped from ~3 to ~1 mg/L in 180 min, presumably due to relatively high levels of organic matter (5.2 ± 0.03 mg/L TOC), whereas there was little to no decay occurred in the Detroit tap water (1.2 ± 0.03 mg/L TOC) over the same time period (Figure 3A). The addition of iron wire to simulate unlined iron pipe corrosion during distribution further reduced chlorine residuals in conditions with both treated Flint River water and Detroit tap water as influents (Figure 3B). However, while some residual was consistently detected in the Detroit tap water effluents after simulated distribution (DET, **DET-cold**, **DET-Enhanced**; 0.5–1 mg/L Cl_2_ after 180-min exposure), treated Flint River water conditions (**FR**, FR-CC, FR-no Fe) generally had no detectable residual (Figure 3B). Condition *FR-NoFe* is not shown in Figure 3B because no iron wire was added to the SDSs for that condition.

While there was variability due to seasonal changes in the source water and variable iron wire corrosion rates throughout the experiment, the mean chlorine concentration (n = 43) after incubation in the SDSs exhibited a general trend of (lowest to highest): *FR-CC* ≈ **FR** < *FR-NoFe* ≈ *DET* ≈ **DET-Cold** < **DET-Enhanced** (Table 1). Based on a Kruskal–Wallis rank sum test, the mean chlorine concentrations across the SDS conditions were significantly different (*p* value < 2 × 10^−16^), while a pairwise post-hoc Tukey test further confirmed specific differences between conditions indicated by a “<” sign in the above trend analysis (all *p* values ≤ 0.009).

Overall, key expectations were also recreated with respect to known trends resulting from water chemistry and corresponding chlorine residual in SDSs effluent. Specifically, the SDSs successfully reproduced chlorine residuals comparable to those during the crisis of 0.28 ± 0.24 mg/L (at Flint city monitoring station 6) [30], compared to levels of 0.26 ± 0.23 mg/L in our treated Flint River water simulation (**FR** condition, Table 1 section C). SDS conditions also successfully simulated pre-crisis (**DET-Cold**) and post-crisis (**DET-Enhanced**) high chlorine, with actual values only 1 mg/L higher than measured during the pre- or post-crisis conditions (Table 1 section C). Both conditions with treated Flint River water and iron present (**FR** and *FR-CC*) occasionally had undetectable chlorine residuals under the conditions tested, whereas *FR-NoFe* and all conditions with Detroit tap water consistently had a measurable chlorine disinfectant residual following simulated distribution, as hypothesized (Figure 1). Iron has been shown to decay chlorine residual in typical drinking waters [31], but the chlorine decay observed in the SDS step was accelerated beyond what is typical due to the corrosivity of the treated Flint River water and lack of corrosion control.

#### 3.1.3. SDSs Iron and Corrosion Control

Known benefits of corrosion control (**FR** vs. *FR-CC*; **FR** vs. *DET*) in terms of hindered iron release and maintenance of higher chlorine residuals in the actual Flint distribution system (Table 1) were not achieved in these simplistic simulations. Based on a prior study [2], the addition of phosphate corrosion control to treated Flint River water reduced iron weight loss by 5.1 times compared to that observed in treated Flint River water without phosphate, while also reducing chlorine decay rates. Further, iron corrosion rates were 8.6 times lower in Detroit tap water with corrosion control versus treated Flint River water without corrosion control, a trend confirmed by our citizen science field sampling throughout Flint in August 2015 versus August 2017 (Figure 1) [3,5]. However, the corrosion control simulation applied to the SDSs in this study did not produce known significant differences in mean effluent iron (i.e., **FR**, *FR-CC*, and *DET*; Table 1 section C). The only condition with relatively low iron in this work was treated Flint River water without any iron present (*FR-NoFe*), in which mean iron was 15.4 ± 19.4 μg/L compared to the 60.5 ± 212 μg/L observed in August 2017 flushed water samples collected in Flint (Table 1).

We were aware that the simple approach applied here would not effectively replicate impacts of iron corrosion control, given that phosphate inhibition of iron corrosion and associated chlorine decay can sometimes require 6–12 months to produce expected benefits even under continuous-flow conditions in water mains, and even longer under more stagnant conditions [32,33]. In this seven-month simulation, the iron was only exposed to the water approximately 6 h each week, which translates into seven days total exposure of iron to the target water over the entire study. Thus, the analysis that follows considers that this particular aspect of the simulation is not representative of what occurred in the field.

### 3.2. Simulated Premise Plumbing Reactors Reproduced Key Water Chemistry Trends of Pre-, During, and Post-Crisis Flint Water

#### 3.2.1. SPPRs Chlorine

After the effluents from the SDSs were transferred to the SPPRs, the 50% water change produced an immediate dilution of chlorine. Beyond dilution, there is an immediate chlorine demand from the combination of aged pipe material, pipe surface area, biofilm, and water within each reactor [24,26]. Notably, all SDS conditions, except **DET-Enhanced**, experienced an immediate chlorine demand within the first 10 min, which exceeded the 50% loss expected from dilution. **FR** and *FR-CC* never received any initial chlorine residual flowing into the SPPRs (Figure 4), whereas *FR-NoFe* retained a low, but detectable, chlorine residual (>0.1 mg/L) for a period of 60–120 min in the PEX SPPRs and 1–10 min in the copper SPPRs (data not shown). Chlorine was reduced in the Detroit tap water SPPRs to below 0.1 mg/L within 30–60 min in both PEX and copper SPPRs, while chlorine was maintained above 0.1 mg/L for up to 120 min in **DET-Cold with** PEX (Figure 4) versus just 10–30 min in the corresponding copper SPPRs. Chlorine residuals in the **DET-Enhanced** conditions after 120 min were 0.92 and 0.38 mg/L in the PEX and copper SPPRs, respectively (Figure 3). In some instances, chlorine was still detectable in **DET-Enhanced** SPPRs after 24 h. As a general rule, when detectable chlorine residual was present in the influent to the SPPRs, levels were higher in the system with PEX after 10 min than in the equivalent system with copper, consistent with the overall hypothesis of this work and our prior research [25] (Figure 1).

Overall, these results illustrate quick and drastic decay of the chlorine disinfectant residual in premise plumbing systems (Figure 4) that added to decay in the distribution systems (Figure 3). The U.S. Environmental Protection Agency (EPA) recommends that a free chlorine disinfectant residual be detectable (often, >0.1 or >0.2 mg/L) in 95% of distribution system samples [34], which has previously been acknowledged not to be adequate for the reduction of *Legionella* in large buildings, single-family homes, or small buildings [35]. The results from the Detroit tap water SPPRs (*DET*, **DET-Cold**, **DET-Enhanced**) demonstrate that the residual was detectable (>0.1 mg/L Cl_2_) after 120 min in the SPPRs only when the disinfectant residual entering much higher than 0.2 mg/L Cl_2_ (Figure 4).

#### 3.2.2. SPPRs Copper

Mean total copper in influent water (Table 1B) to all SPPRs was consistently < 15 μg/L and mean effluent copper from PEX reactors was consistently < 100 μg/L, in accordance with the assumption that the only source of copper was traces from plumbing used to collect well-flushed raw water samples in the field. However, SPPRs containing copper pipe consistently produced effluent with total mean copper concentrations > 700 μg/L (Table 1 section C) and were statistically higher than the copper concentrations from the PEX SPPRs effluent (*p* value = 2 × 10^−16^). Further, effluents from SPPRs receiving treated Flint River water, simulating Flint water during the crisis (**FR**), contained higher total copper than each of the corresponding conditions representing Detroit water (**DET-Cold**, **DET-Enhanced**, *p* values < 4 × 10^−5^), consistent with the lack of copper corrosion control during this time period. Thus, the laboratory simulation successfully reproduced the trends in copper levels characteristic of pre-/during and post-crisis conditions in Flint, where mean, 5th, and 95th percentiles of first draw copper during the crisis were approximately three times higher than post-crisis (Table 1 section A,C).

### 3.3. Legionella pneumophila Response to Simulation of Water Chemistry and Premise Plumbing Material

After 101 days of SPPRs acclimation to the SDSs water and the cross-inoculation period, culturable *L. pneumophila* numbers were greatest in SPPRs receiving treated Flint River waters, particularly the PEX condition (Figure 5A). Among all SPPRs containing PEX material, *L. pneumophila* CFU/mL were significantly higher in treated Flint River water-sourced (Figure 5A) compared to Detroit-sourced (Days 25–175; *p* < 0.05; Figure 5C) water. This demonstrated the main hypothesized effect of treated Flint River water being more conducive than Detroit tap water to maintaining viable *L. pneumophila,* at least in the absence of copper pipe (Figure 1).

Throughout the study, *L. pneumophila* persisted at low numbers in the copper SPPRs fed with treated Flint River waters, but at levels significantly lower than in the PEX SPPRs (*p* value = 0.03). In particular, the copper SPPRs receiving *FR-NoFe* influent water sustained little to no culturable *L. pneumophila* beyond 75 days (Figure 5B). During our field sampling at the height of the summer 2015 LD outbreak, the pH was 7.0 in Flint homes, in which case the higher acidity likely caused much higher levels of Cu^+2^ in premise plumbing [3,12,14] than in this study at a pH of 7.8. Together, the results from the treated Flint River water copper SPPRs at a pH 7.8 suggest that, under conditions of corrosive influent water (including the *FR-CC* water as evidenced by chlorine decay tests; Figure 3B), the elevated copper concentrations can enhance reduction of *L. pneumophila*, consistent with the overarching hypotheses of this study (Figure 1).

The Detroit tap water conditions provide a simulation of what occurred before the city of Flint switched to the treated Flint River water (April 2014), and after they switched treated Flint River water back to the Detroit municipal water supply on 16 October 2015. Initial culturable counts of *L. pneumophila* declined under all Detroit tap water conditions within the first month (Figure 5C,D), which was consistent with our field data [1,3] and the corresponding drop in LD incidence after switching back to Detroit water [5]. The loss of culturable *L. pneumophila* was greatest in PEX SPPRs for all three Detroit tap water SDS conditions, with culturable *L. pneumophila* falling below detection by Day 25, with a single exception (Figure 5C). Interestingly, *L. pneumophila* fared better in Detroit SPPRs containing copper pipe relative to those containing PEX pipe material, consistent with a previous study in the same Detroit tap water PEX reactors with no chlorine and additional mixing [12]. Plate counts in Detroit SPPRs containing copper remained near 1 CFU/mL from Day 25 until the end of the experiment. This suggests the additional chlorine demand and reduced chlorine levels caused by the presence of copper can actually increase growth of *Legionella* as hypothesized (Table 1 and Figure 1). Notably, the colony counts in the **DET-Enhanced** condition dropped below detection after just 100 days (Figure 5D), illustrating that extra chlorine can overcome the demand exerted by copper and more effectively control *L. pneumophila*. The presence of orthophosphate in Detroit tap water also would have reduced toxicity of the *L. pneumophila* to copper as reported in earlier studies [12,24,36].

#### 3.3.1. Isolate Analysis

To gain insight into whether a single strain or mixture of strains of *Legionella* persisted under the various conditions, 56 representative isolates collected from the SPPRs on Days 0, 47, 82, 175, and 210 were subject to genotypic screening by PCR. Interestingly, it was observed that *L. pneumophila* survived through the end of the experiment across all simulations, except **DET-Enhanced** (Appendix A). Of the *L. pneumophila* strains recovered from SPPRs fed with treated Flint River water by Day 150, 56/56 were characterized as serogroup 1. By Day 210, **FR** and all other water conditions supporting *Legionella* were confirmed to contain a mixture of serogroup 1 and non-serogroup 1 *L. pneumophila*, based on PCR detection of the wzm gene. The ability to multiple serogroups of *L. pneumophila* to persist under the various conditions of this experiment suggests that the trends observed in response to the water conditions employed in this study were robust across multiple serogroups.

#### 3.3.2. Chlorine Disinfectant

Chlorine is by far the most widely implemented and relied upon secondary disinfectant residual applied in the U.S. and around the world. Previous studies have indicated that concentrations >0.5 mg/L are sometimes sufficient for limiting detectable levels of *Legionella* in large building plumbing water systems [37,38]. As demonstrated above, once added to SPPRs, chlorine concentrations immediately decreased and were often no longer detectable after 60 min following a water change. Further, in all SPPRs, except those receiving **DET-Enhanced** water, chlorine concentrations decreased below 0.5 mg/L within the first 10 min. The persistence of culturable *Legionella* in all conditions, except the **DET-Enhanced**, essentially proves that adequate disinfection was not achieved for the other 10 conditions

#### 3.3.3. Pipe Material: PEX and Copper

Comparing copper versus PEX pipe materials further illustrated the potential for premise plumbing conditions to mediate the effects of the distribution system water chemistry. Copper is of particular interest because of its known antimicrobial properties towards *L. pneumophila* [28,39,40,41]. Ironically, the lack of corrosion control, which triggered higher iron and lower chlorine residuals and exacerbated *Legionella* problems in some portions of the Flint distribution system during the crisis [3], also released high levels of copper that might have helped to control *Legionella* growth in some buildings and homes. In this study, regardless of influent water sources, copper SPPRs displayed rapid initial loss of culturable *L. pneumophila* subsequent to feeding the SDS waters. However, low colony counts persisted in most SPPR conditions containing copper at the higher pH, representative of the summer 2014 outbreak, for nearly six months. The initial reduction of *Legionella* CFUs in copper SPPRs may have occurred through limited antimicrobial properties of aged copper at the relatively high pH employed in this study, as observed by others [12,24,42,43].

As *Legionella* are facultative intracellular bacteria, they are capable of residing in biofilms and replicating in more than 20 species of amoebae [44,45]. In harsh environments, such as the surface of copper pipe material or variable disinfection levels of chlorine, existing biofilms may serve as a protective environment for *Legionella* to shelter from disinfectants [26,43,45]. However, this study suggests that such protective mechanisms can be overcome by higher chlorine residual and contact time. This is demonstrated by the **DET-Enhanced** copper condition, which included increased chlorine and corrosion control agent and contained no culturable *L. pneumophila* from Day 50 forward (with the exception of a single colony obtained from a 50 mL sample of water on Day 82) (Figure 5D).

Interactions between copper and influent chlorine residuals may provide an environment that allows for the persistence of *Legionella* in premise plumbing, especially if corrosion control maintains Cu^+2^ below thresholds controlling *Legionella* [14,19,46]. A previous study determined that disinfectant potential of free chlorine can be affected by the age of copper pipe material [28]. In both treated Flint River water and Detroit tap water, the presence of aged copper reduced the capacity of the SPPRs to maintain measurable free chlorine residuals (Figure 5B), but the presence of orthophosphate corrosion control may have allowed *Legionella* to persist at higher levels in *DET* or **DET-Cold** water, whereas the corresponding condition with higher copper in **FR** helped reduce *Legionella*.

### 3.4. Experimental Conditions: Hypothetical Effects of Stagnation, Chlorine, and Elevated pH

Comparing results of the current study to those obtained to those from a companion study using the same SPPRs, but with continuous mixing, slightly lower pH, and no disinfectants (Figure 6) [12] can provide insight into overarching effects of experimental conditions selected for this study. Notably, *L. pneumophila* in the treated Flint River water with PEX was 2.5-log higher in the previous study. We hypothesize that this is attributable to continuous mixing versus stagnation, since the small pH change is not expected to be influential for PEX, and chlorine levels delivered to this reactor from the SDSs in this work were consistently undetectable. A much smaller increase of 0.4–1.0 log was observed for treated Flint River water copper conditions in the prior study versus this work, mostly likely due to the lower pH significantly enhancing bacteriostatic impacts of cupric ion. In any case, the higher *L. pneumophila* with more mixing is consistent with prior results in recirculating versus non-recirculating systems [17,19,47,48] in which there were warm temperatures and low disinfectant residuals. In Detroit tap water conditions, comparable conditions between the two studies consistently led to non-detectable *L. pneumophila*, except for copper pipe with both mixing and corrosion control, where relatively low levels of *L. pneumophila* persisted.

## 4. Conclusions

The findings of this study are consistent with the understanding that *L. pneumophila* is not uncommon in municipal water flowing into buildings [49], but a range of water chemistry and premise plumbing conditions and disinfectant residuals can prevent their proliferation. Persistent disinfectant, including chlorine, is known to be a critical factor in reducing *Legionella* risk [35,37,39], and a recent study confirmed predicted associations between low levels of chlorine in Flint’s distribution system and observed incidence of LD [5]. Consistent with our prior in-field observations [1,3], we further demonstrate under controlled laboratory conditions the importance of considering interactive effects with flow and pipe materials, particularly with respect to relative water corrosivity and influence on residual chlorine levels, in keeping *Legionella* levels low. Indeed, many individual factors can act as “two-edged swords” in terms of their net effect of controlling versus enhancing *Legionella* growth, depending on the status of other factors. For example, copper pipe achieves its best antimicrobial efficacy without corrosion control, but absence of corrosion control also leads to elevated iron and depleted chlorine residual, which in turn enhance *Legionella* growth. Such interactive effects can help explain why prior studies reported relatively low levels of *Legionella* in single family homes, which tend to have greater stagnation and more copper in water from copper service lines and plumbing, compared to large multi-story buildings during the Flint Water Crisis.

## Figures and Tables

**Figure 1 pathogens-09-00730-f001:**
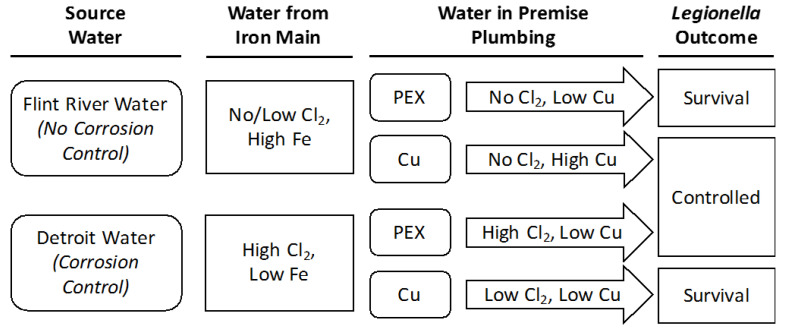
Experimental framework and specific hypotheses for this study. Corrosivity of the municipal water supply influences levels of chlorine and iron in the water delivered to the premise plumbing. Premise plumbing materials, cross-linked polyethylene (PEX) or copper (Cu), further alter the water chemistry and overall propensity for *Legionella* to be controlled or to survive. Corrosion of copper and iron will consume free chlorine, whereas plastic materials have little or no chlorine demand. Corrosive water will also release soluble copper ions from copper pipe, especially in stagnant premise plumbing conditions. Elevated levels of either copper or chlorine can control *Legionella*.

**Figure 2 pathogens-09-00730-f002:**
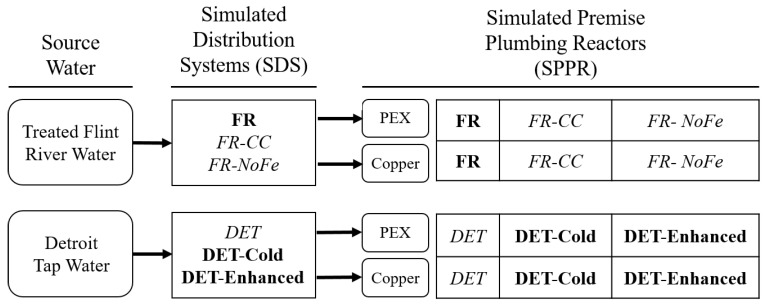
Experimental design from source water to simulated distribution systems (SDSs) to simulated premise plumbing reactors (SPPRs). Source water was treated and stored in 10–30 L batches until fed to SDSs. Each SDS condition was set up in a flask containing 330 mL of source water, an iron wire (except for *FR-no Fe*), a stir bar, and 3.0 or 3.5 mg/L chlorine. Bold conditions were designed to replicate scenarios found before (**DET-Cold**), during (**FR**), or after the Flint Water Crisis (**DET-Enhanced**). Conditions in *italics* were designed to simulate hypothetical scenarios if corrosion control had been implemented or if water had not flowed through unlined iron pipe. After the SDSs simulation was completed, the water was fed to corresponding SPPRs containing either PEX (n = 3) or copper (n = 3). The total number of SPPRs was 36.

**Figure 3 pathogens-09-00730-f003:**
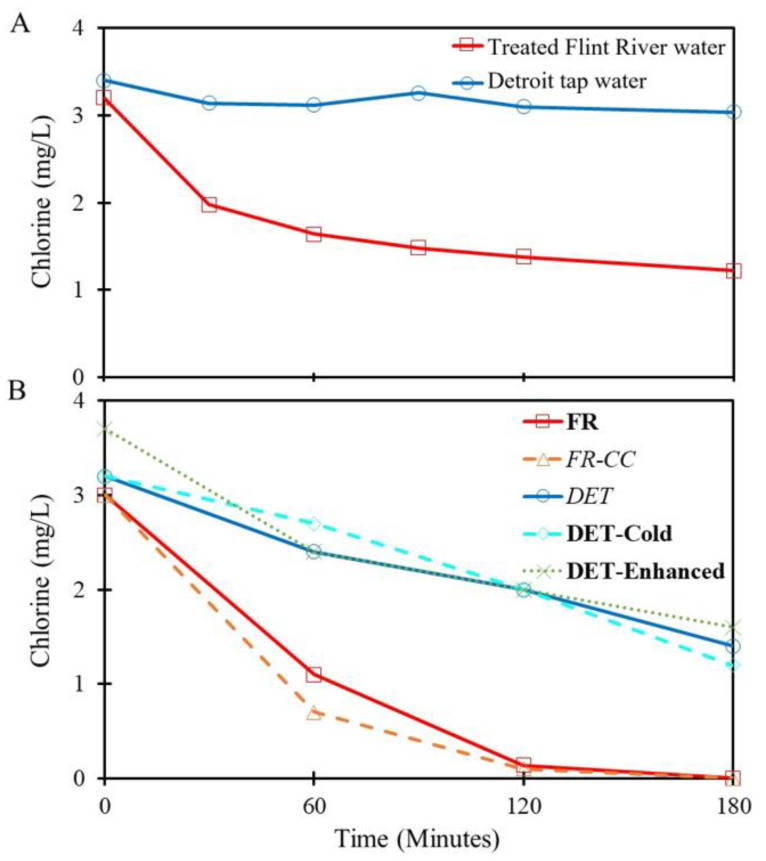
Representative Cl_2_ decay in source water and simulated distribution systems (SDSs). (**A**) Control experiment of chlorine decay of treated Flint River and Detroit tap water in non-reactive glass reactors without iron. (**B**) Representative results in different SDSs conditions: **FR**, treated Flint river water aged with iron wire; *FR-CC,* treated Flint River water with added corrosion control and aged with iron wire; *DET*, Detroit tap water aged with iron wire; **DET-Cold**, Detroit tap water incubated at cooler temperature with iron wire; **DET-Enhanced**, Detroit tap water with additional corrosion control and initial elevated chlorine levels.

**Figure 4 pathogens-09-00730-f004:**
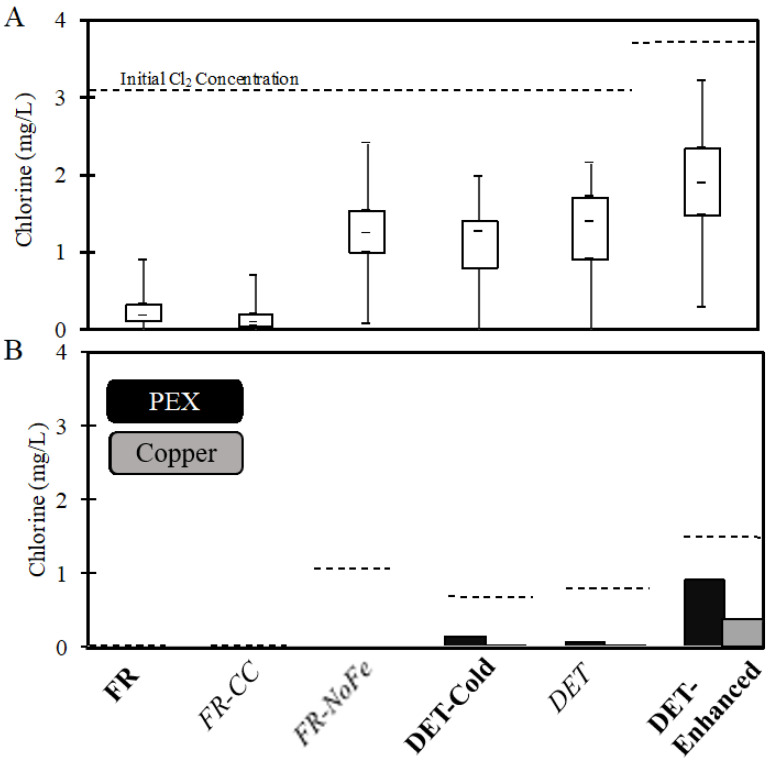
Chlorine residuals (**A**) after 3 h contact time in the simulated distribution systems (SDSs) and (**B**) 120 min after the effluent from the SDSs were fed to the simulated premise plumbing reactors (SPPRs) (50% fresh SDSs water with 50% remaining SDSs following incubation in the SPPRs the previous cycle). Dashed lines indicate the calculated initial chlorine level added for each water or reactor type. Bars represent the maximum and minimum, the upper and lower bounds of the box are the first and third quartiles, and the median is indicated by the internal dash. The detection limit was 0.02 mg/L.

**Figure 5 pathogens-09-00730-f005:**
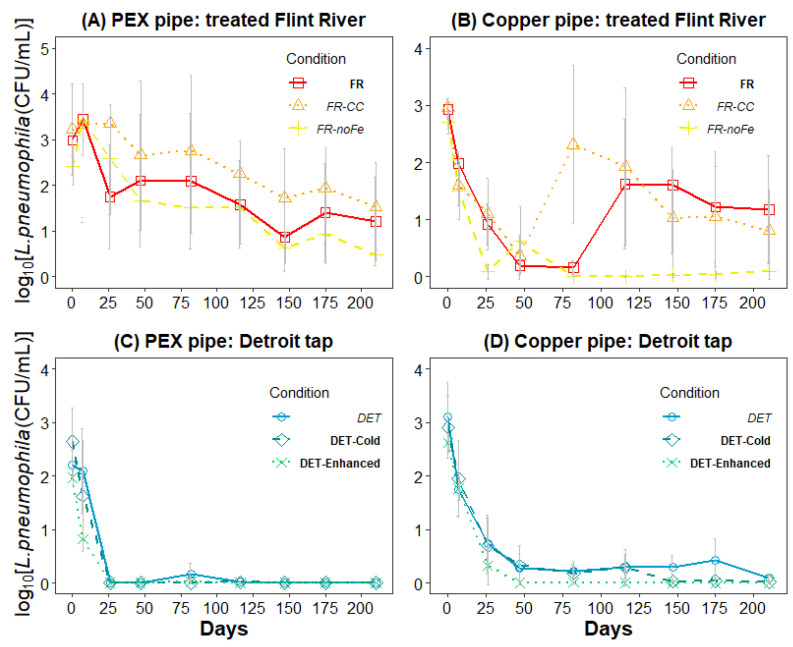
The effects of water source, distribution conditions, and pipe material on culturable *L. pneumophila* in simulated premise plumbing reactors (SPPRs). Effluent log transformed average *L. pneumophila* numbers (CFU/mL) and standard deviations from SPPRs receiving simulated distribution system (SDS) water corresponding to: (**A**) treated Flint River water with PEX pipe coupons; (**B**) treated Flint River water with copper pipe coupons; (**C**) Detroit tap water with PEX pipe coupons; and (**D**) Detroit tap water with copper pipe coupons. All SPPRs influent waters were spiked to an initial target concentration of 3 mg Cl_2_/L and aged three hours under completely-mixed conditions at 23 °C in the presence of an iron wire (SDS step), except *FR-NoFe* conditions, which had no iron wire, and **DET-Cold** condition, which was incubated at 17 °C. *FR-CC* additional orthophosphate corrosion control agent added at 1 mg/L, **DET-Enhanced** additional CL_2_ added at 3.5 mg/L and orthophosphate at 2.5 mg/L.

**Figure 6 pathogens-09-00730-f006:**
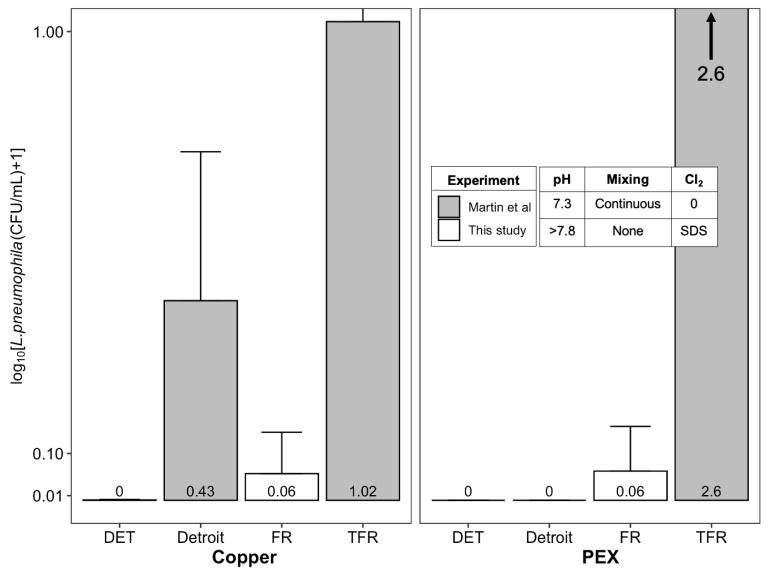
Comparison of culturable *L. pneumophila* (CFU/mL) from control reactors in this study (Day 210 samples only) compared to Martin et al. (sampled at one year). Each bar in the plot represents the average of triplicate reactors with the error bars showing the standard deviation. The Detroit condition contained the same influent as DET, with no iron and 2.5 mg/L of orthophosphate. The treated Flint River (TFR) condition contained the same influent as FR except without the iron aging step.

**Table 1 pathogens-09-00730-t001:** Relevant Flint Water Crisis data compared to simulated treatment, distribution, and premise plumbing conditions.

A: Flint Water Crisis Field Data *				B: Simulated Distribution System Influent ^						C: Simulated Distribution System (T, Cl_2_, Fe) or Simulated Premise Plumbing Reactor (Cu) Effluent ^+^					
Condition	Pre-Crisis	Crisis	Post-Crisis	**DET-Cold**	**FR**	**DET-Enhanced**	*FR-CC*	*FR-NoFe*	*DET*	**DET-Cold**	**FR**	**DET-Enhanced**	*FR-CC*	*FR-NoFe*	*DET*
Temp (°C)	20 ± 2.2	23 ± 2.1	18 ± 2.4	18	22	22	22	22	22	18 ± 1.4	22 ± 1.3	22 ± 1.3	22 ± 1.3	22 ± 1.3	22 ± 1.3
Chlorine (mg/L Cl_2_)	0.50 ± 0.19	0.28 ± 0.24	0.38 ± 0.16	3.10 ± 0.20	3.10 ± 0.21	3.80 ± 0.19	3.10 ± 0.21	3.10 ± 0.21	3.10 ± 0.20	1.28 ± 0.23	0.26 ± 0.23	1.87 ± 0.70	0.17 ± 0.20	1.27 ± 0.49	1.07 ± 0.52
Flushed Total Iron (μg/L)	UNK	208 (21–340)	42.4 (0–130)	92.1 (80–110)	15.4 (2.0–47)	92.1 (80–110)	15.4 (2.0–47)	15.4 (2.0–47)	92.1 (80–110)	1300 (490–2500)	2300 (590–5200)	900 (360– 2000)	3100 (690–9500)	15.4 (2.0–47)	2400 (710–6000)
Total Copper (μg/L)	UNK	129 (14–380)	46 (3.0–140)	12 (5.7–20)	8.0 (4.1–15)	8.0 (5.7–20)	8.0 (4.1–15)	8.0 (4.1–15)	12 (5.7–20)	962 (510–1390)	2380 (1790–3120)	510 (170– 1100)	1200 (520–2000)	650 (530–850)	880 (640–1400)

* Section (**A**): Representative chemical mean, ± standard deviation, (5–95 percentile range where available) for peak Legionnaires’ Disease (LD) months of June–September for the indicated stage of the Flint Water Crisis. Representative distribution system temperature data are reported in Rhoads et al. (2017), chlorine data are from monitoring station 6, iron data are from citizen science sampling of flushed water from the same 150 homes in August 2015 (crisis) and August 2017 (post-crisis), and copper water crisis data are from a subset of first draw samples from homes that records indicate had at least partial copper service lines (n = 79). UNK = unknown. ^ Section (**B**): Same parameters as in Section (A) were measured in the influent to the SDSs mean (5th–95th percentile) (n = 10 samples over a 10-month period) (Ambient laboratory set point reported for temperature data). Bold conditions SDSs were designed to simulate actual conditions found during the crisis. Conditions in italics were designed to simulate hypothetical scenarios. “DET” conditions received Detroit tap water influent. “FR” conditions received treated Flint River influent. ^+^ Section (**C**): Mean and standard deviation of temperature and chlorine and mean (5th–95th percentile) of iron effluent from SDSs (i.e., influent to the SPPRs) and mean (5th–95th percentile) of copper in the effluent of copper pipe SPPRs.

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
