# Peer review of "Interactive Effects of Copper Pipe, Stagnation, Corrosion Control, and Disinfectant Residual Influenced Reduction of Legionella pneumophila during Simulations of the Flint Water Crisis"

_pathogens, 2020, doi:10.3390/pathogens9090730_

Round 1

Reviewer 1 Report

Summary: This study set out to experimentally examine changes in water chemistry that may have contributed to the Flint Legionella outbreaks that occurred after switching municipal water sources from the Detroit water supply to the Flint River and the impact of piping material and corrosion control treatment. Overall, the study seems well designed, but the abbreviation of the study conditions adds to reader confusion and reduces interpretability. The conclusions of the study seem interesting but are not well contextualized in the literature overall beyond previous Flint-specific studies from the authors.

Major comments:

The title of the study does not seem to adequately reflect the findings or at least leads to confusion. “Stagnant conditions, corrosion control, and trace disinfectant residuals enhanced reduction of Lp”. Stagnant conditions and trace disinfectant residual should increase Lp not “enhance reduction” as noted in lines 249-251 and elsewhere. “Lack of corrosion control exacerbated Legionella problems” lines 278-281. But also, ”…presence of orthophosphate corrosion control may have allowed Legionella to persist at higher levels in the DET or DET-Cold water” Lines 301-302.

There are a lot of specific conditions and abbreviations for them throughout the manuscript. Some do not seem as well defined as others so it is very hard to follow what each abbreviation means. In Table 1 each condition should be explained in a line for easy reference throughout the manuscript. Since the methods are at the end of the manuscript instead of before the results/discussion the abbreviations are often not explained at their first appearance. It sounds like there are multiple abbreviations for the same test conditions. Please clarify these throughout.

Overall, the merged results and discussion had very little discussion.

The conclusion about single family homes vs. multi-story buildings having low levels of Legionella is not really fleshed out throughout the manuscript. I would like more information/detail about this to be included

Minor comments:

Figure 1 appears well before it is cited in the text of the manuscript. Also, there is a strange extra black line and arrow on the bottom right “Survival” box.

Lines 97-99: the phrasing of this sentence is a little confusing. It lists SDSs, TFR, and DTW and then two chlorine levels and ‘respectively’, then goes on to explain DTW chlorine level. Perhaps leave DTW out of the first sentence fragment then just describe after the Table 1B reference. Or list all 3 if there are supposed to be 3 and then the DET-Enhanced is a different condition than the DTW?

What “DET” “DET-Cold”, “DET-Enhanced”, “FR-CC”, or “FR-NoFe” are defined too late in the manuscript. Could you provide reference to those conditions upon first mention in the results/discussion section?

Line 118- Figure 6 is referenced before Figure 2? Perhaps these need to be renumbered or moved?

Line 123- notes “insignificant decay” but lists no p value. Are there statistics to support the claim this was or was not significant or a different word should be chosen.

Figure 2- Are FR and TFR water the same? And DET and DTW? It seems strange to have two different acronyms for the same test condition.

Figure 3- Hash pattern for “copper” makes the box describing what the color means hard to read.

Lines 183-184: how was a 2-3 day water change for 50% volume chosen? Is this from a reference or specific experience in the field. Please cite or explain.

Line 263: “to persist” appears a smaller font than the rest of the line

Why are Legionella counts in dL not a standard measure such as L or mL?

Line 363: Should be section 3.2.1. not 2.2.1.

Line 399 has an extra indent

Why were culture results <20 CFU considered non-quantifiable?

Table S1. Consider making the “X” for positive a “+”.

Reviewer 2 Report

This is a well-written manuscript with very interesting findings. This article simulated the Flint water crisis using SDSs and SPPRs to identify the effects of water chemistry and other factors (e.g., pipe material) on the community-wide LD outbreaks which happened during the summers of 2014 and 2015. The study is well designed and conducted, and the statistics are adequate. I just have a few minor comments for the authors. Below are my specific comments:

  • Lines 38-39: Please cite appropriate references if possible.
  • Why not move the Figure 1 to the materials and methods section?
  • Line 115: It should be Section C.
  • Line 295: What could be the possible reason for that colonization on Day 82?
  • I believe that moving the materials and methods section before the results and discussion section would be helpful for readers to connect the experimental design with the results and vice-versa.
